# Impact of the Recipient’s Pre-Treatment Blood Lymphocyte Count on Intended and Unintended Effects of Anti-T-Lymphocyte Globulin in Allogeneic Hematopoietic Stem Cell Transplantation

**DOI:** 10.3390/cells12141831

**Published:** 2023-07-12

**Authors:** Alexander Nikoloudis, Veronika Buxhofer-Ausch, Christoph Aichinger, Michaela Binder, Petra Hasengruber, Emine Kaynak, Dagmar Wipplinger, Robert Milanov, Irene Strassl, Olga Stiefel, Sigrid Machherndl-Spandl, Andreas Petzer, Ansgar Weltermann, Johannes Clausen

**Affiliations:** 1Ordensklinikum Linz—Elisabethinen, Department of Internal Medicine I: Hematology with Stem Cell Transplantation, Hemostaseology and Medical Oncology, 4020 Linz, Austria; 2Medical Faculty, Johannes Kepler University, 4020 Linz, Austria

**Keywords:** GVHD, ALC, ATLG, HSCT

## Abstract

**Background:** In allogeneic hematopoietic stem cell transplantation (HSCT), Anti-T-Lymphocyte Globulin (ATLG) may be used for the prevention of severe graft-versus-host disease (GVHD). ATLG targets both the recipient’s lymphocytes and those transferred with the graft. Assuming an inverse relation between the recipient’s absolute lymphocyte count (ALC) and exposure of remaining ATLG to the graft, we aim to evaluate the impact of the recipient’s ALC before the first ATLG administration on the benefits (prevention of GVHD and GVHD-associated mortality) and potential risks (increased relapse incidence) associated with ATLG. **Methods:** In recipients of HLA-matched, ATLG-based HSCT (n = 311), we assessed the incidence of acute GVHD, GVHD-related mortality and relapse, as well as other transplant-related outcomes, in relation to the respective ALC (divided into tertiles) before ATLG. **Results:** The top-tertile ALC group had a significantly increased risk of aGVHD (subhazard ratio (sHR) 1.81; [CI 95%; 1.14–2.88]; *p* = 0.01) and aGVHD-associated mortality (sHR 1.81; [CI 95%; 1.03–3.19]; *p* = 0.04). At the highest ATLG dose level (≥45 mg/kg), recipients with lowest-tertile ALC had a trend towards increased relapse incidence (sHR 4.19; [CI 95%; 0.99–17.7]; *p* = 0.05, n = 32). **Conclusions:** ATLG dosing based on the recipient’s ALC may be required for an optimal balance between GVHD suppression and relapse prevention.

## 1. Introduction

Allogeneic hematopoietic stem cell transplantation (HSCT) is a potentially curative treatment for various hematological malignancies and non-malignant disorders. However, a major complication associated with HSCT is graft-versus-host disease (GVHD). Prophylactic treatment with immunosuppressive drugs, typically a combination of a calcineurin inhibitor and an antimetabolite, is an essential component of HSCT to reduce the incidence and severity of GVHD. However, even with a two-drug combination, acute and chronic GVHD still constitute significant clinical problems, particularly following HSCT from unrelated and/or HLA mismatched donors. In order to increase the efficacy of the GVHD prophylactic backbone regimen, T-cell-directed serotherapy using either Anti-T-Lymphocyte Globulin (ATLG, Grafalon R), Anti-Thymocyte Globulin (ATG, Thymoglobulin R), or the CD52 antibody alemtuzumab [1] has been successfully introduced in unrelated HSCT [2,3,4], and also in matched related HSCT [5]. Both ATLG and ATG are polyclonal antibody preparations, obtained from the serum of rabbits immunized with a human T-lymphoblastic (Jurkat) cell line (ATLG), or primary human thymocytes (ATG). However, since there have been concerns raised about overdosing on ATLG [6], particularly in the context of low lymphocyte counts of the recipient at the time of ATLG initiation [7], the optimal dosage of ATLG that should be used in an individual HSCT recipient remains a subject of ongoing research. The current standard of care is to administer a fixed dose of ATLG based on the recipient–donor relationship and the recipient’s body weight [8].

While for ATLG, two studies, i.e., the US randomized ATLG trial [7] and a large retrospective analysis from Essen, Germany [9], have suggested an association of a lower absolute lymphocyte count (ALC) on the first day of serotherapy with an increased relapse risk or inferior overall survival, respectively, a number of studies have systematically addressed the role of the recipient’s ALC in ATG-treated recipients. Research findings indicate that the recipient’s ALC and ATG dosages may exhibit a potential relationship concerning the incidence of aGVHD [10,11,12]. In addition, Marsh et al. [13] reported an association between pre-dose ALC and post-transplant serum levels of alemtuzumab, which were frequently observed to be in excess of levels required for the prevention of severe GVHD. The rationale behind this approach is that the ALC reflects the number of T-lymphocytes in the recipient, which are the target of ATLG, ATG, and alemtuzumab. A higher recipient’s ALC may lead to the consumption of a higher proportion of the respective T-cell-directed serotherapeutic agent, lowering its overall levels remaining in circulation to interact with the grafted T-cells. Therefore, a higher dose of serotherapy may be needed in recipients with high ALC at the time of the first serotherapy infusion to achieve sufficient depletion of the grafted T-lymphocytes for effective GVHD prevention. On the other hand, a low recipient ALC may result in excessive circulating serotherapy levels remaining on day 0, which may subsequently increase the risk of relapse by hampering adequate post-transplant immune reconstitution (Figure 1).

In this study, we aim to evaluate the role of the recipient’s ALC at the time of the first ATGL (Grafalon) infusion on endpoints related to insufficient control of alloimmunity (aGVHD, cGVHD, NRM, aGVHD-associated mortality) and the most important endpoint related to excessive immunosuppression, namely, relapse of the underlying malignancy. Further, the impact of the recipient’s pre-ATLG ALC on overall survival will be addressed to explore whether ALC may guide optimal dosing of ATLG in HLA-matched HSCT.

## 2. Materials and Methods

This retrospective monocentric study from an Austrian EBMT center (EBMT-CIC 594-Linz) included all consecutive HSCTs performed with G-CSF mobilized peripheral blood stem cells (PBSC) or bone marrow (BM) and applied standard GVHD prophylaxis (i.e., CSA/MTX or CSA/MMF), augmented with ATLG (Grafalon), between 27 December 2001 and 18 December 2018 with a follow-up until 5 April 2022 (n = 311). The majority of HSCTs were performed after the year 2010 (71.1%). All patients provided written informed consent to personal data management for scientific purposes. The informed consent form and process for personal data management were approved by national authorities and the institutional review board.

The impact of the pre-ATLG ALC on the following endpoints was analyzed: incidence of non-relapse mortality (NRM), acute GVHD (aGVHD), chronic GVHD (cGVHD), aGVHD-associated mortality, relapse incidence, and overall survival (OS).

Table 1 summarizes the patient, disease, and transplant characteristics for the cohort.

In detail, post-grafting immunosuppression consisted of CSA (initial trough level 200–300 ng/mL, with tapering between day 100 and day 180 in the absence of active GVHD), combined with either MTX (15 mg/m^2^ on day +1, 10 mg/m^2^ on days +3, +6, +11; day +11 MTX was omitted in the case of relevant toxicities such as active infection or severe mucositis), or with MMF (MMF dose was either 1000 mg 2 times per day, 1000 mg three times per day, or 12.5 mg/kg three times per day; MMF was routinely continued until day +35 or until day +56). ATLG dosing mainly depended on individual conditioning protocols that were currently active and the donor–recipient relationship. Until 2014, ATLG dosing mainly depended on individual conditioning protocols that were currently active, and ATLG was predominantly used in matched unrelated HSCT. From 2015 on, ATLG was routinely used both in matched sibling and matched unrelated HSCT, and dosing was mainly based on the donor–recipient relationship. Total ATLG dose varied between 15 mg/kg and 60 mg/kg (median 30 mg/kg). ATLG infusions were divided into 2–4 days, with the latest ATLG infusion on day −2 or −1.

Conditioning regimens were categorized as myeloablative (MAC) or reduced-intensity (RIC) according to published criteria [14].

### 2.1. Definitions and Grading

Overall survival (OS) represents the duration from the transplantation procedure until death from any cause, or until the last recorded follow-up if no death has occurred. Progression-free survival (PFS) is identified as the span from transplantation until the point of disease recurrence or death from any cause. Non-relapse mortality (NRM) is characterized as the occurrence of death without signs of primary disease activity or a preceding hematological relapse.

aGVHD-associated mortality was defined as any non-relapse mortality associated with a previous or active case of grade 2–4 acute graft-versus-host disease. Disease risk was classified as follows. Acute leukemia in its first complete remission, low-risk MDS or MPN, or non-malignant diseases were considered as low-risk factors. Intermediate-risk factors included acute leukemia in its second complete remission or intermediate-risk MDS or MPN. High-risk factors encompassed acute leukemia exhibiting active disease at the time of transplantation, high/very high-risk MDS or MPN, and any disease that recurred after a previous allogeneic stem cell transplantation. For the risk stratification of MDS and MPN, the currently active international prognostic scoring systems were utilized at the time of diagnosis and transplantation. Lymphomas and plasma cell disorders were categorized in accordance with the revised disease risk index [15]. The grading of acute GvHD was conducted based on the (modified) Glucksberg criteria [16]. Chronic GvHD was categorized following the Seattle criteria, being labeled as either “limited” or “extensive”, or based on the NIH 2005 consensus criteria, which includes designations of “mild”, “moderate”, or “severe” [17,18].

### 2.2. Statistical Methods

For all multivariate analyses, the following covariates were initially considered: the recipient’s age (age_r), donor age (age_d), sex-mismatched female donor (sex_mm), HLA mismatch (mm), unrelated versus related donor (relation), disease risk (DR), donor and recipient CMV serostatus (cmv_d), conditioning intensity (cond), ATLG dose in mg/kg body weight (ATLG_dose), methotrexate-based GVHD prophylaxis versus other (MTX_bsd), a time variable for transplantation date (time_TX), graft source (source), minor AB0 mismatch (min_AB0mm), major AB0 mismatch (maj_AB0mm), and infused cell dose in 10^8^ cells per Kg bodyweight (mnc).

To assess the impact of high and low ALC in the peripheral blood before the first ATLG administration, we divided the cohort into tertiles (thirds) defined by their ALC, resulting in a low-ALC tertile with an ALC ≤ 0.04 Giga/L (n = 109), a high-ALC tertile ≥ 0.14 Giga/L (n = 107), and an intermediate tertile with ALC > 0.04 and <0.14 Giga/L. Since borderline ALC values can be overlapping the calculated tertiles (e.g., 0.05 Giga/L), the 3 resulting cohorts do not exactly have the same size. The decision to subdivide the cohort into three subgroups by ALC was based on the intention to achieve groups of similar and statistically sufficient size. Moreover, similar cut-offs were used in other studies using Thymoglobulin [11,19].

The log-rank test and Kaplan–Meier statistical methods were applied for single-variable analyses of overall survival and progression-free survival. All single-variable outcomes, including survival probabilities and cumulative incidences, are presented for a three-year period following HSCT. For multivariable analyses, the statistical model used was the Cox proportional hazard model. 

In the assessment of non-relapse mortality (NRM) and mortality associated with aGVHD, relapse was treated as a competing risk, and the same applied in reverse. Gray’s test was used to compare the cumulative cause-specific incidence functions for single-variable analyses of these competing risks. For multi-variable analysis, the Fine and Gray regression method was employed.

We hypothesized that a low-tertile ALC was predictive for an excessive ATLG effect (i.e., a potentially increased relapse incidence), while a high-tertile ALC predicts GVHD-related outcomes (NRM, aGVHD-associated NRM, aGVHD 2–4, aGVHD 3–4, cGVHD). For overall survival, both cut-offs were tested, hypothesizing that an intermediate ALC may represent an optimal condition, balancing the risks and benefits of anti-T-cell serotherapy, resulting in a net survival benefit. The variable selection process involved backward selection based on the Akaike Information Criterion (AIC). The final model was chosen based on the AIC, and the reported results of the target variable (e.g., ALC cut-off tertiles) represent either the final model or, in case the target variable was removed during the selection process, the last model in which it was included. To study the effect of ATLG dose and ALC levels on progression-free survival, we used a multivariate Cox regression model as described above. In order to examine whether the effects of ATLG vary across different levels, an interaction term was included in the analysis. To account for potential nonlinearity or skewness in the ALC levels, a logarithmic transformation (Log(x + 1)) was applied in order to facilitate the examination of the effects of ATLG across different levels. In our hazard ratio models, an interaction term between ALC and ATLG dose signifies that the impact of ALC on the risk of an event depends on the ATLG dose. If the interaction term is >1, the risk effect of ALC increases more strongly as the ATLG dose increases. If it is =1, the effect of ALC on the risk does not change with different ATLG doses. If it is <1, the risk effect of ALC reduces as the ATLG dose increases.

The R-packages survival, riskRegression, cmprsk, cowplot, survminer, finalfit, gt, rms, and ggplot2 were used for the statistical analyses and plots [20].

## 3. Results

### 3.1. Determination of the Recipient’s ALC before First ATLG Administration

The median pre-serotherapy ALC was 0.07 Giga/L, ranging from 0.00 Giga/L to 13.40 Giga/L. Based on tertiles, we defined three ALC groups: a low-ALC tertile with ALC ≤ 0.04 Giga/L (n = 109), a high-ALC tertile with ALC ≥ 0.14 Giga/L (n = 107), and an intermediate tertile with ALC > 0.04 Giga/L and < 0.14 Giga/L (n = 95). The outcomes of the multivariate analyses for the major endpoints are summarized in Table 2.

### 3.2. Acute GVHD Grade III-IV, aGVHD-Associated Mortality, and Chronic GVHD

The univariate analysis showed a trend towards higher incidence of aGVHD grade 3–4 in HSCT recipients with a top-tertile ALC (ALC high) on the first day of ATLG administration, compared to those of the low/intermediate-ALC group (cumulative incidence at 3 years: 28% versus 21%; *p* = 0.08). After adjusting for potential confounders in the multivariate analysis, the ALC high group had a significantly higher risk of aGVHD grade 3–4 (subhazard ratio (sHR) 1.81; [CI 95%; 1.14–2.88]; *p* = 0.01). The impact of a high ALC was even stronger when high-risk transplantations, i.e., patients with an advanced disease stage or uncontrolled disease at transplant, were excluded from the multivariate model (sHR 2.08; [CI 95%; 1.02–4.24]; *p* = 0.04).

Similarly, the univariate analysis showed a higher incidence of aGVHD-associated mortality in the ALC high group (Figure 2B; cumulative incidence at 3 years: 20% versus 14%; *p* = 0.20). After adjusting for other factors in the multivariate analysis, the ALC high group had a significantly higher risk of aGVHD-associated mortality (1.81; [CI 95%; 1.03–3.19]; *p* = 0.04). As above, this effect was even more pronounced in the non-high-risk disease setting (SHR 2.15; [CI 95%; 0.99–4.67]; *p* = 0.05).

In contrast, the interaction between the transformed ALC variable and disease risk had no significant impact on aGVHD-associated mortality and aGVHD-associated mortality.

A significantly higher cumulative incidence for moderate/severe chronic GVHD could be seen for the ALC high population (cumulative incidence at 3 years: 25% versus 16%; *p* = 0.04), which was recovered as a trend in the multivariate model (sHR 1.60; [CI 95%; 0.97–2.62]; *p* = 0.06).

### 3.3. Overall Survival, Non-Relapse Mortality, Incidence of Relapse, and Progression-Free Survival

The univariate analysis found no notable distinctions in overall survival rates for patients within the ALC groups (Figure 2A; *p* = 0.57). Likewise, the multivariate analysis showed an adjusted hazard ratio (aHR) of 1.19; [CI 95%; 0.8–1.78] for the trend over the tertiles, confirming the absence of a significant impact of the ALC on overall survival (*p* = 0.38).

Patients in the ALC high group had no significantly higher rate of non-relapse mortality compared to the lower two tertiles by univariate analysis (cumulative incidence at 3 years: 23% versus 22%; *p* = 0.49). After adjusting for confounding factors in the multivariate analysis, the ALC high group had a trend towards a higher risk of NRM (sHR 1.41; [CI 95%; 0.87–2.27]; *p* = 0.16).

There was a weak trend towards a higher relapse rate among patients with low ALC on the day of the first ATLG administration compared to recipients in the upper two tertiles (Figure 2C; cumulative incidence at 3 years: 40% versus 32%; *p* = 0.12). However, this trend could not be confirmed by multivariable analysis (sHR 1.06; [CI 95%; 0.71–1.58]; *p* = 0.76).

Like the overall survival, progression-free survival showed no difference for ALC in the trend over the tertiles (sHR 0.99; [CI 95%; 0.69–1.43]; *p* = 0.95).

### 3.4. Interaction between ATLG Dose and ALC

The interaction between the recipient’s ALC (measured in cells × 10^9^/liter) and the total ATLG dose (per kg body weight) was a significant predictor for progression-free survival (aHR 0.91; [CI 95%; 0.85–0.97]; *p* = 0.004). This interaction term represents the change in hazard for a one-unit increase in the ALC and a one-unit increase in the ATLG dose. Specifically, it suggests that for each additional unit of ALC and ATLG dose, the hazard ratio for progression-free survival decreases by approximately 9%. The relative risk of PFS failure (relapse or death from any other cause) associated with increasing ATLG doses is higher in recipients with lower ALC than in those with higher ALC. Conversely, the relative risk of PFS failure associated with higher ALC was higher in individuals with lower ATLG doses than in individuals with higher doses. As shown in Figure 3, a low dose of ATLG (15 mg/kg body weight) was associated with a lower progression-free survival with increasing ALC, while a high dose of ATLG (60 mg/kg body weight) was associated with a lower progression-free survival with decreasing ALC based on the multivariate model.

### 3.5. Subgroup Analyses of Conditioning Intensity, Donor Relationship, and ATLG Dose Levels

The impact of a high ALC on the incidence of NRM was significant in RIC transplantations (sHR 2.39; [CI 95%; 1.19–4.81]; *p* = 0.02; n = 145), but not present in MAC transplantations (sHR 1.42; [CI 95%; 0.73–2.76]; *p* = 0.30; n = 166). An interaction between the transformed ALC variable and conditioning intensity was not significant in NRM (sHR 4.37; [CI 95%; 0.58–33.04]; *p* = 0.15).

While in unrelated donor transplantations, high ALC was associated with a higher risk of NRM (sHR 1.50; [CI 95%; 0.91–2.47]; *p* = 0.11; n = 240), no impact was observed in HLA-matched sibling transplants (sHR 1.51; [CI 95%; 0.24–9.47]; *p* = 0.66; n = 71). The interaction between the transformed ALC variable and donor relation was not significant.

For the relapse incidence, low ALC had no significant impact in any of the above subgroups.

ATLG dose was split into a lower-dose group (≤30 mg/kg; n = 178) and a higher-dose group (≥35 mg/kg; n = 133) for stratification analyses regarding the impact of different ALC levels on different dosages of ATLG.

In transplantations with a lower ATLG dosage, high ALC was not significantly associated with an increased incidence of aGVHD grade 3–4 (sHR 1.64; [CI 95%; 0.89–3.02]; *p* = 0.11). Also, in transplantations with a higher ATLG dosage, there was no significant impact in aGVHD grade 3–4 associated with a high ALC (1.87; [CI 95%; 0.85–4.14]; *p* = 0.12). A significantly increased risk of aGVHD-associated mortality was seen for the ALC high group in transplantations with lower ATLG dose levels (sHR 2.51; [CI 95%; 1.06–5.91]; *p* = 0.03) but not at higher ATLG dose levels (sHR 2.02; [CI 95%; 0.83–4.88]; *p* = 0.12).

An interaction between the transformed ALC variable and bodyweight-based ATLG dose failed to be significant and was observed as a trend for acute GVHD (sHR 0.90; [CI 95%; 0.79–1.01]; *p* = 0.07) and NRM (0.89; [CI 95%; 0.8–1.00]; *p* = 0.06).

The transformed ALC variable failed to show significance for progression-free survival with a low dose level (HR 2.21; [CI 95%; 0.48–10.15]; *p* = 0.30) and a high dose level (HR 0.73; [CI 95%; 0.34–1.57]; *p* = 0.43).

To further analyze the observed effect, we exclusively analyzed recipients diagnosed with either AML or ALL for aGVHD 3–4. This analysis of AML and ALL confirmed the significantly higher risk of aGVHD in the “high ALC” group (SHR 2.06; [95% CI; 1.10–3.87]; *p* = 0.02; n = 197).

## 4. Discussion

Upon administration a few days prior to HSCT, ATLG targets both the recipient’s lymphocytes and the lymphocytes transferred with the graft. A higher recipient ALC may therefore require higher doses of serotherapy for effective GVHD prevention, while a low ALC might result in excessive ATLG blood levels remaining for the graft, which could lead to an increased relapse incidence due to impaired post-transplant T-cell reconstitution.

Importantly, the majority of previous studies on the impact of the recipient’s ALC in transplants using in vivo T-cell depletion refer to ATG Thymoglobulin [10,11,12,19,21,22,23,24,25] or alemtuzumab [13,23] as serotherapy, whereas the evidence for ATLG (Grafalon) remains limited [7,9]. Furthermore, it is important to recognize that the above studies utilized different serotherapy dosing and/or timing [7,9,10], which may limit the applicability of their findings to a setting in which the first day of ATLG administration is predominantly day −3 (interquartile range, day −4 through day −2), as in our cohort. Given these differences, the present study provides an important complement to the existing evidence. 

In our cohort, a high ALC was strongly associated with an increased incidence of aGVHD and GVHD-related mortality. These findings are consistent with previous studies on ATG, which suggest that ALC may be a predictor of aGVHD risk and that a higher dose of ATG may be required to achieve adequate T-lymphocyte depletion in recipients with elevated ALC [10,11,12]. An association between pre-transplant ALC and both ATG pharmacokinetics and post-HSCT immune reconstitution has been described before in studies on ATG and alemtuzumab [13,26].

Several previous studies on ATG or ATLG [7,9,10,11,12,19,22,23,25] have reported an impact of ALC on overall survival, most likely due to the association of a high aGVHD incidence with high ALC levels [10,11,12]. 

Despite increased aGVHD-associated mortality in recipients with high ALC in our cohort, overall survival was not significantly decreased. This suggests that a reduced risk of relapse, although not significant in the overall cohort, may partly offset the negative impact of high ALC on aGVHD-associated mortality. Indeed, subgroup analyses revealed only a trend towards a higher relapse incidence (sHR 4.19; [CI 95%; 0.99–17.7]; *p* = 0.05) in recipients received high dose levels of ATLG (≥45 mg/kg, n = 32). In agreement with these findings, a low ALC was previously reported to be linked to lower relapse-free survival [24] and increased relapse-related mortality [10]. Study cohorts with predominantly high or very high disease risk and unrelated donor transplantations [24] or high dosages of serotherapy [10] are in agreement with our observation of a low ALC being associated with an increased relapse risk in the above subgroup. Further research is needed to determine a more suitable cut-off for low ALC that would better reflect its interaction with ATLG and the relapse incidence.

Not in all studies was ALC described to impact the outcome in ATG-based transplantations [21]. A recent pediatric study analyzed the ATG dose in relation to the respective recipient’s ALC, revealing that a lower ATG/ALC ratio was associated with a higher risk of GVHD, a lower risk of treatment-related mortality, and with better overall survival [25]. In empirical post hoc analyses of our cohort, the predictive value of the ALC/ATLG ratio, applying weight-based dosing, was an inferior outcome predictor than the ALC itself. Nevertheless, our finding of a significant interaction between ATLG dose and ALC is in agreement with Grasso’s observation. As previously reported for the interaction between ALC and ATG dose for overall survival [19], a similar interaction could be found between ATLG dose and ALC for progression-free survival in our study (Figure 3). 

Our findings are in agreement with previous studies suggesting that individualizing the dose of ATLG (or ATG) on the basis of the recipient’s ALC may improve transplant outcomes. In particular, a high ALC at the time of serotherapy initiation is predictive of an increased aGVHD risk, and in these recipients, higher doses of ATLG (or ATG) may be necessary to achieve adequate T-lymphocyte depletion. While our data reveal a significant association of an elevated ALC with an increased incidence of aGVHD and aGVHD-related mortality, the fact that in our study, a low ALC only weakly correlated with an increased relapse incidence may be the result of a merely moderate ATLG dosing in the majority of the investigated transplants. 

The major limitation of the present study is the long observation period of the study, which we tried to meet with a variable for the transplant era in the multivariate models. Since the algorithm of ATLG use and dosage has changed over time from a rather protocol-driven to a more risk-adapted approach, we have further included the respective risk factor, i.e., the recipient/donor relationship, in the multivariate models. This was a single-center retrospective trial that needs to be confirmed by future prospective and multi-center studies.

Despite these limitations, our study adds substantially to the previous evidence on the clinical importance of the recipient’s lymphocyte count at the onset of anti-T-cell serotherapy, since the majority of previous studies investigated ATG (Thymoglobulin)- and non-ATLG (Grafalon)-based HSCT. Only two studies [7,9] addressed this issue in the context of ATLG (Grafalon), and the respective cohorts differed from the present one with respect to the ATLG dose, being higher in the full cohort prospectively studied by Soiffer et al., and partly higher in the study by Turki et al. Also, both of the above studies consistently applied MTX-based post-grafting immunosuppression. Since both high-dose ATLG and MTX may increase the risk of relapse [6,7,27], these differences are important, and optimal ALC cut-offs may therefore not be transferable from one setting to another. 

## 5. Conclusions

Individualization of the ATLG (or ATG) dose should take into account not only whether the donor is related or unrelated, but also the recipient’s ALC on the first day of serotherapy, besides other established GVHD risk factors such as HLA-C killer cell immunoglobulin receptor (KIR) ligand status and donor–recipient sex mismatch [6], and the type of post-grafting immunosuppression [27]. Further studies are required to establish the optimal extent of dose adjustment in recipients with very high or very low ALC, in order to achieve an optimal balance of aGVHD prevention and relapse prevention in HST.

## Figures and Tables

**Figure 1 cells-12-01831-f001:**
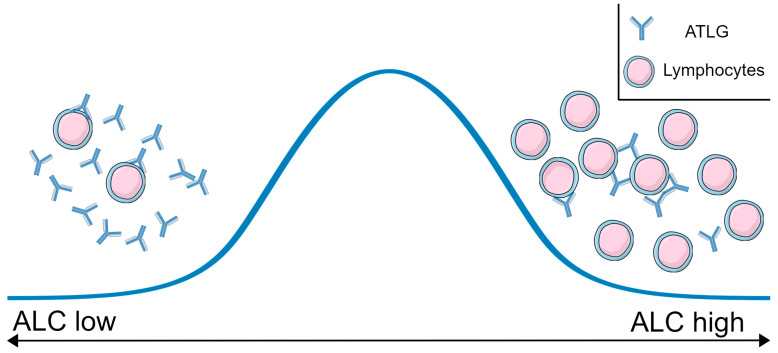
Graphical representation of the underlying hypothesis tested.

**Figure 2 cells-12-01831-f002:**
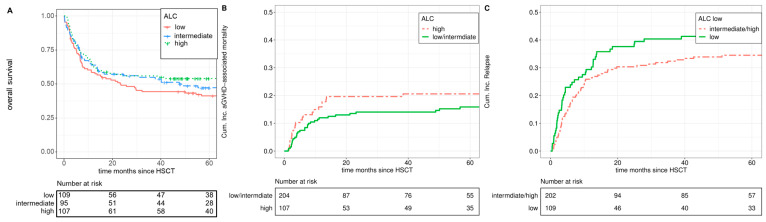
ALC by different endpoints. (**A**) OS for ALC low (red), ALC intermediate (blue), ALC high (green). (**B**) GVHD-related mortality for ALC high (red), ALC low/intermediate (green). (**C**) Relapse incidence for ALC low (green), ALC intermediate/high (red).

**Figure 3 cells-12-01831-f003:**
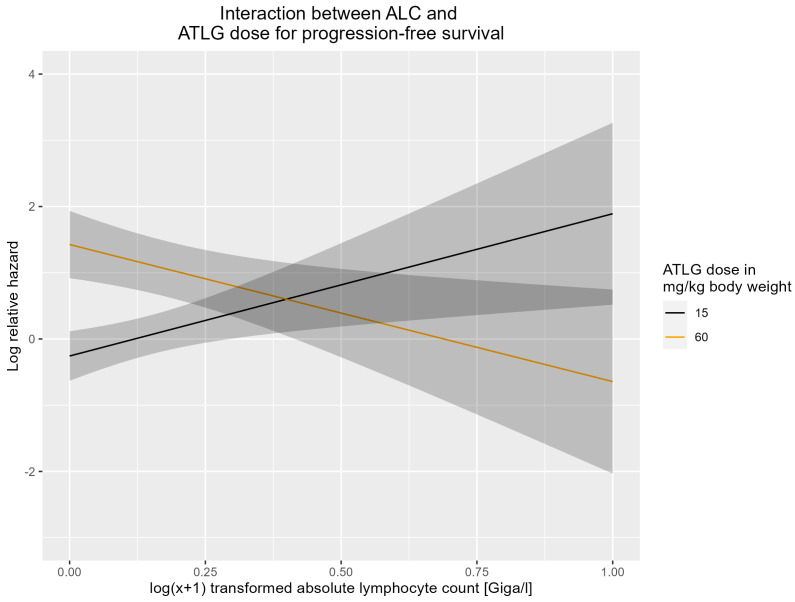
Graphical representation of the interaction between the log(x + 1) transformed ALC variable and low dose of ATLG (15 mg/kg body weight; black) and high dose of ATLG (60 mg/kg body weight; yellow) based on the multivariate progression-free survival model, given as log relative hazard with 95% confidence intervals.

**Table 1 cells-12-01831-t001:** Summary of the cohort characteristics. ^1^ Absolute lymphocyte count; ^2^ myeloablative conditioning; ^3^ reduced-intensity/non-myeloablative conditioning; ^4^ calcineurin inhibitor/mycophenolate-mofetil or other; ^5^ calcineurin inhibitor/methotrexate; ^6^ female donor to male recipient; ^7^ peripheral blood stem cells.

Cohort Characteristics		
Factor	Group	Overall
n		311
age donor (median [range])		35.00 [18.00, 70.20]
age recipient (median [range])		50.60 [18.00, 73.00]
ATG type (%)	ATLG	311 (100.0)
ATLG dose (median [range])		30.00 [10.00, 60.00]
ALC (median [range]) ^1^		0.07 [0.00, 13.40]
ALC group (%)	lower third (tertile)	109 (35.0)
	middle third (tertile)	95 (30.5)
	upper third (tertile)	107 (34.4)
CMV IgG status donor (%)	no	192 (61.7)
	yes	119 (38.3)
CMV IgG status recipient (%)	no	150 (48.2)
	yes	161 (51.8)
disease risk (%)	low	104 (33.4)
	intermediate	81 (26.0)
	high	126 (40.5)
conditioning (%)	MAC ^2^	166 (53.4)
	RIC ^3^	145 (46.6)
HLA matching (%)	10/10 matched	239 (76.8)
	9/10 matched	72 (23.2)
GVHD prophylaxis (%)	CNI/MMF or other ^4^	233 (74.9)
	CNI/MTX ^5^	78 (25.1)
donor type (%)	matched related	71 (22.8)
	unrelated	240 (77.2)
sex mismatched (%) ^6^	no	243 (78.1)
	yes	68 (21.9)
graft source (%)	bone marrow	16 (5.1)
	PBSC ^7^	294 (94.9)
diagnosis (%)	MDS or MPN	57 (18.3)
	AML	146 (46.9)
	lymphoma or myeloma	47 (15.1)
	ALL	51 (16.4)
	nonmalignant	10 (3.2)

**Table 2 cells-12-01831-t002:** Summary of results of multivariate analyses for the major endpoints. * marks the covariates remaining in the respective model.

Multivariate analyses of the impact of absolute lymphocyte count on study endpoints		
Endpoint	(s)HR	*p*-value
overall survival * time_TX, ATLG_dose, relation, age_d, age_r, DR, mm, MTX_bsd, source, maj_AB0mm		
ALC low/intermediate/high	1.19	0.38
progression-free survival * time_TX, DR, ATLG_dose, relation, cond, age_d, age_r, DR, cmv_d, cmv_r, mm, MTX_bsd, sex_mm, source, maj_AB0mm, min_AB0mm, mnc		
ALC low/intermediate/high	0.99	0.95
non-relapse mortality * time_TX, ATLG_dose, DR, cmv_d, cmv_r, age_r, mm, relation, source, maj_AB0mm, mnc		
ALC high	1.41	0.16
aGVHD-associated mortality * time_TX, lctest_Dummy, age_r, mm, relation, min_AB0mm, mnc		
ALC high	1.81	**0.04**
Relapse * time_TX, ATLG_dose, age_d, age_r, DR, cmv_d, cmv_r, cond, mm, MTX_bsd, relation, sex_mm, maj_AB0mm, min_AB0mm, mnc		
ALC low	1.06	0.76
aGVHD 2–4 * cond, DR, mnc, age_r, time_TX, ATLG_dose, relation		
ALC high	1.24	0.20
aGVHD 3–4 * DR, mm, source, mnc		
ALC high	1.81	**0.01**
cGVHD moderate/severe or extensive * ATLG_dose, DR, relation, min_AB0mm		
ALC high	1.60	0.06

## Data Availability

The data presented in this study are available upon request from the corresponding author. The data are not publicly available due to restrictions concerning privacy or ethical considerations.

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
