# Peer review of "Impact of the Recipient’s Pre-Treatment Blood Lymphocyte Count on Intended and Unintended Effects of Anti-T-Lymphocyte Globulin in Allogeneic Hematopoietic Stem Cell Transplantation"

_cells, 2023, doi:10.3390/cells12141831_

Round 1

Reviewer 1 Report

This manuscript by Nikoloudis et al. explores the relationship between the recipient's absolute lymphocyte count (ALC) before the administration of anti T lymphocyte globulin (ATLG- Grafalon) in 311 patients who underwent allogeneic hematopoietic cell transplantation (HCT) between 2001 and 2018 and its impact on GVHD, GVHD-related mortality, and relapse incidence. This study addresses an important clinical question in HCT and complements recent publications on this topic by others. 

They demonstrate that a high ALC was strongly associated with an increased incidence of aGVHD and GVHD-related mortality which is consistent with previous studies using ATG and suggest that ALC may be a predictor of aGVHD risk and that a higher dose of ATG may be required to achieve adequate T-lymphocyte depletion in recipients. They also found an association (P=0.05) between the highest ATLG dose level recipients with lowest-tertile ALC had a trend towards increased relapse.

The authors acknowledge the limitations of this study which include different dose schedules of ATLG and retrospective study over a long18-year time span.

Overall, the manuscript is well-written and easy to follow and contributes to the evolving literature on the relationship of recipient ALC and dosing of T-lymphodepleting serotherapy.

Author Response

Reviewer 1

This manuscript by Nikoloudis et al. explores the relationship between the recipient's absolute lymphocyte count (ALC) before the administration of anti T lymphocyte globulin (ATLG- Grafalon) in 311 patients who underwent allogeneic hematopoietic cell transplantation (HCT) between 2001 and 2018 and its impact on GVHD, GVHD-related mortality, and relapse incidence. This study addresses an important clinical question in HCT and complements recent publications on this topic by others.

They demonstrate that a high ALC was strongly associated with an increased incidence of aGVHD and GVHD-related mortality which is consistent with previous studies using ATG and suggest that ALC may be a predictor of aGVHD risk and that a higher dose of ATG may be required to achieve adequate T-lymphocyte depletion in recipients. They also found an association (P=0.05) between the highest ATLG dose level recipients with lowest-tertile ALC had a trend towards increased relapse.

The authors acknowledge the limitations of this study which include different dose schedules of ATLG and retrospective study over a long18-year time span.

Overall, the manuscript is well-written and easy to follow and contributes to the evolving literature on the relationship of recipient ALC and dosing of T-lymphodepleting serotherapy.

We would like to express our gratitude for your time and dedication, as well as for providing feedback on our manuscripts.

Reviewer 2 Report

The authors found that high levels of ALC in preconditioned recipients were associated with increased risk of grades III-IV acute GVHD and increased acute GVHD-related mortality in transplant recipients in which ATLG was used as conditioning regimen. Furthermore, it was shown that PFS tended to decrease as the ALC increased when the amount of ATLG was low and vice versa when the amount of ATLG was high. These findings may indicate that ALC in the preconditioned recipients may be a factor that determines the optimal amount of ATLG.

1. The methods should indicate the rationale for the determination of the ATLG dose.

2. Statistical methods may include factors that influence GVHD development, such as the number of cells transfused and the presence or absence of major/minor ABO mismatch.

3. In ALC low in Fig 2BC, the types of No and Yes lines are interchanged, which should be fixed.

4. In Table 2, all confounders used should be presented. Also, risk factors for developing grades II-IV acute GVHD should be demonstrated.

5. It is recommended that the results of multivariate analysis by ATLG dose be shown so that the results in Figure 3 can be clearly shown.

6. ATLG should affect donor as well as recipient lymphocytes. It is desirable to demonstrate the effect of donor lymphocyte counts. Or the authors should clarify this point.

7. Due to the small number of cases and high cohort diversity, I would like the authors to demonstrate whether similar results can be obtained in analyzes restricted to AML and ALL.

Round 2

Reviewer 2 Report

The authors well responded to all comments.